# DynaNav: Dynamic Feature and Layer Selection for Efficient Visual Navigation

**Jiahui Wang**[1]    **Changhao Chen**[2]

[1]College of Design and Engineering, National University of Singapore
[2]PEAK-Lab, The Hong Kong University of Science and Technology (Guangzhou)
wjiahui@u.nus.edu    changhaochen@hkust-gz.edu.cn

## Abstract

Visual navigation is essential for robotics and embodied AI. However, existing foundation models, particularly those with transformer decoders, suffer from high computational overhead and lack interpretability, limiting their deployment in resource-tight scenarios. To address this, we propose **DynaNav**, a **Dyna**mic Visual **Nav**igation framework that adapts feature and layer selection based on scene complexity. It employs a trainable hard feature selector for sparse operations, enhancing efficiency and interpretability. Additionally, we integrate feature selection into an early-exit mechanism, with Bayesian Optimization determining optimal exit thresholds to reduce computational cost. Extensive experiments in real-world-based datasets and simulated environments demonstrate the effectiveness of DynaNav. Compared to ViNT, DynaNav achieves a $2.26\times$ reduction in FLOPs, 42.3% lower inference time, and 32.8% lower memory usage, while improving navigation performance across four public datasets.

## 1 Introduction

Visual navigation is a fundamental capability for robotics and embodied AI, enabling autonomous agents to perceive, interpret, and navigate complex 3D environments based on visual inputs [1–3]. Its applications span real-world scenarios, such as delivery and logistics, as well as virtual domains, including gaming and simulation. By bridging perception and action, visual navigation plays a crucial role in intelligent systems. Recently, there has been growing interest in developing foundation models for visual navigation [4–10, 1]. ViNT [5] is a notable example that learns from large-scale egocentric observations using transformer layers on CNN-extracted features, demonstrating strong generalization across robotic platforms and environments. NoMad [6] further builds on this by incorporating a diffusion policy and a goal-masking mechanism. PixNav [9] utilizes textual heuristics and large language models(LLMs) to explore zero-shot possibility. However, these approaches, particularly those relying on deep neural architectures such as transformer decoders, introduce significant computational overhead, posing challenges for edge deployment where efficiency is paramount.

Robotic applications demand greater efficiency than large cloud-based models. As the trend toward efficient foundation models continues [11, 12], reducing the computational burden of visual navigation models is a key challenge. Additionally, existing models function as "black boxes," raising concerns about interpretability. As humans and intelligent agents increasingly coexist, explainability becomes essential. These challenges lead to two critical research questions:

- *Is it necessary to activate all transformer layers for every navigation scenario?*

- *Which features are most important in the decoding process, and can we identify the most salient regions or pixels for navigation?*

39th Conference on Neural Information Processing Systems (NeurIPS 2025).

Humans do not always activate all neurons for visual tasks [13]; rather, the brain dynamically recruits resources based on task complexity. Inspired by this, we propose that visual navigation models should adopt dynamic inference mechanisms, selectively utilizing features and neural layers based on scene complexity. In simpler scenarios, the model should rely on fewer features and layers for efficient computation, whereas in more complex tasks, it should allocate additional resources to ensure accurate decision-making.

To this end, we propose **DynaNav**, a highly efficient **Dyna**mic Visual **Nav**igation framework that adaptively selects relevant features and neural layers based on visual observations. Our approach employs a trainable hard feature selector to create sparse representations, enabling computationally efficient sparse operations at the feature level. This dynamic feature masking not only lowers computational overhead but also improves the understanding of which regions more relevantly influence the inference of visual navigation models, thereby enhancing explainability. Additionally, we introduce an early-exit strategy for deep Transformer layers by integrating feature selection into the early-exit mechanism, improving stability and computational efficiency. After training the decoder, Bayesian Optimization determines optimal early-exit thresholds. During inference, if a layer's feature meets its threshold, computation terminates early, significantly reducing overall computational cost. Extensive experiments on real-world datasets and in simulated environments demonstrate the effectiveness of our proposed DynaNav. Compared to ViNT [5], DynaNav achieves a $2.26\times$ reduction in FLOPs, 42.3% lower inference time, and 32.8% lower memory usage while improving navigation performance across four public datasets. To the best of our knowledge, this is the first work to introduce dynamic network mechanisms to visual navigation models. To sum up, the main contributions of our work can be summarized as follows:

- We propose DynaNav, a highly efficient and effective dynamic neural model for visual navigation, introducing a novel feature and layer selection strategy to improve efficiency without compromising performance.
- We integrate sparse feature selection into the early exit mechanism, improving the stability and success rate of dynamic layer inference, while the visualized mask enhances the interpretability of the navigation decision process.
- Extensive experiments and simulations demonstrate that DynaNav achieves more than twice the efficiency while maintaining comparable success rates.

## 2 Related Work

### 2.1 End-to-end Visual Navigation

Nowadays, conducting robot learning from diverse datasets to obtain a general model is becoming more and more popular [14–16]. Nonetheless, current approaches rely on real-world data, which is usually costly to obtain, lacks generalization, and is highly coupled with specific robot settings that are hard to transfer to different platforms [17, 18]. Instead, our paper follows the paradigm of learning navigation behavior from data collected across multiple different real-world robotic systems [19, 20, 4] while focusing on training a foundation model that can be adapted for various downstream tasks in zero-shot or with small amounts of data. To this end, models like RT-1, I2O, and GNM [21, 4, 22] provide useful insights that study broad generalization across environments and embodiments for robots deployed in real-world settings. GNM [4] demonstrates policy learning from heterogeneous RGB datasets but focuses on the singular task of reaching image goals in the zero-shot setting. ViNT [5] trains an effective visual navigation policy that can solve a range of downstream tasks, such as navigating to GPS goals [23], goal images [24], and skill-conditioned driving [25]. Building upon extensive prior work in visual navigation, ViNT combines two key elements: it uses topological graphs to keep track of how spaces are connected in the environment while employing trained policies to handle the detailed movement controls [26–29, 7, 30]. Recently, NoMaD [6] boosted the navigation task in previously unseen environments with goal masking techniques and diffusion policy.

### 2.2 Dynamic Network and Early Exit

Dynamic networks [31–33] tend to optimize models that can modify their architecture or parameters based on the input during the inference process. There are many techniques to achieve a dynamic

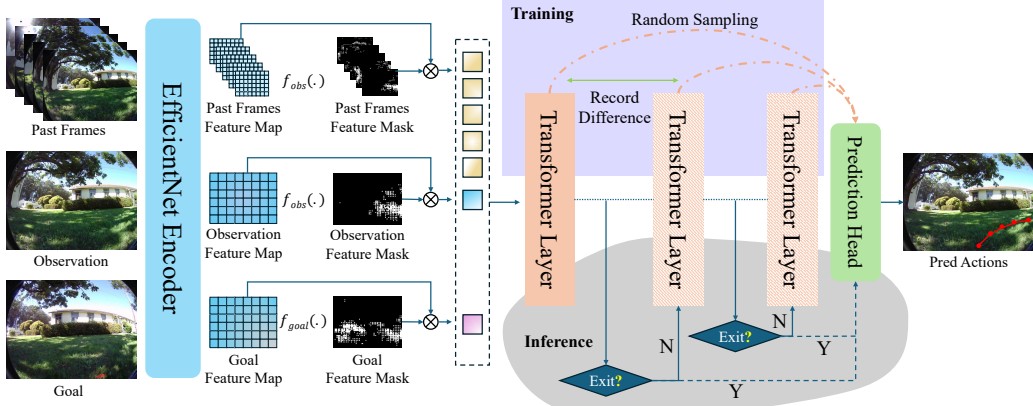

Figure 1: The architecture of our DynaNav framework. DynaNav employs two encoder instances of the same architecture: one processes the concatenated observation and historical frame sequence, while the other extracts features from the early-fused combination of the current observation and goal image. Two independent feature selectors generate masks for observations and the goal, which are then tokenized and fed into a Transformer decoder. Training incorporates stochastic early-exit triggers at intermediate decoder layers. During inference, a decision step at each layer evaluates optimized thresholds to determine whether early exit conditions are met.

network depth-wise and width-wise. For instance, layer-skipping [34], neurons-skipping [35], and low rank approximation (LoRA) [36]. Moreover, some dynamic networks focus on adjusting the shape and value of weights adaptively during the inference, such as deformable convolution [37], dynamic filter network [38], etc.. Among these techniques, early exiting gained popularity because of the prevalent Transformer-based model, which fits the inherent architecture with stacked blocks.

**Early exiting** is a depth-wise dynamic method for halting forward propagation at a certain layer based on intermediate predictions. This technique has been well explored in both computer vision [39–44], language processing [45–51], and multimodality [52, 53]. One challenge in implementing early-exiting models lies in finding a suitable metric to decide when to make an intermediate prediction. Commonly, metrics like probability confidence [41] or entropy value [47] are employed in traditional vision tasks. Some research also pointed out the possibility of using learning-based early exit, which relies on a trained network [54, 46, 55, 56]. Recent research [32, 34] has extended early exiting to the LLMs, which treat the autoregressive task as a classification subgoal.

In this work, we are the first to utilize the idea of early exiting on an end-to-end visual navigation model. We further develop the current method [11] with the integration of sparse feature selection to the Bayesian optimization process in obtaining the desirable metric.

## 3   Dynamic Visual Navigation Model

### 3.1   Framework Overview

To enable efficient and effective visual navigation, we propose DynaNav, a dynamic navigation pipeline illustrated in Figure 1. Unlike previous end-to-end models with static network inference, DynaNav integrates a dynamic feature selector and an early-exit mechanism, reducing computational costs while enhancing explainability and robustness. DynaNav begins with an EfficientNet backbone [57] to extract features from RGB image sequences. Building on ViNT [5], we introduce a feature selector module that generates masks before feature processing in the Transformer decoder, allowing for sparse computation. Additionally, we implement a dynamic Transformer decoder, enabling predictions at intermediate layers to improve efficiency. Finally, Bayesian optimization [11] determines the optimal early-exit thresholds for our jointly trained model, further minimizing computational overhead.

**Feature Extraction.** We chose EfficientNet-B0 as our visual encoder due to its innovative compound scaling method, which optimally balances network width, depth, and resolution. Mathematically, the encoder processes an input sequence consisting of consecutive visual observations $\mathbf{o}_i$, where $i \in [t - p, t]$, along with a goal image $\mathbf{o}_s$. Each observation is first mapped to a latent space representation by the encoder, denoted as $\psi(\mathbf{o}_i) \in \mathbb{R}^{H \times W \times C}$, where $\psi(\cdot)$ represents the network, and $H, W, C$ correspond to the height, width, and number of channels in the feature map. To enhance the connection between the current observation and the goal, we adopt an early fusion strategy. Specifically, the goal image is processed separately by another EfficientNet instance, producing a feature representation denoted as $\phi([\mathbf{o}_t; \mathbf{o}_s]) \in \mathbb{R}^{H \times W \times C}$, where $[;]$ represents concatenation. The detailed hyperparameter settings are provided in Section B.1.

**Dynamic Feature Selector.** However, when the embedded feature map is large [11], computing such a tensor in a transformer incurs significant computational costs [58], which limits the navigation model's efficiency. Moreover, not all pixels in the observations and goal are essential; some redundant pixels can be ignored to improve processing efficiency [59]. To address this, we introduce a dynamic hard feature selector that generates a mask to filter out pixels with minimal relevance to the final prediction.

**Transformer Decoder.** After the feature selection process, a transformer decoder $\mathbf{D}$ is employed to extract contextual features for action prediction, as illustrated in Figure 1. The stacked multi-head self-attention (MHSA) layers continuously refine the contextual information of visual tokens. Formally, we define the intermediate token as:

$$\mathbf{x}_i = D_{1:i}\left([\mathbf{m}_{t-p:t} * \psi(\mathbf{o}_{t-p:t}); \mathbf{m}_s * \phi([\mathbf{o}_t; \mathbf{o}_s])]\right), \tag{1}$$

where $\mathbf{m}_{t-p:t}$ represents the generated masks for $\mathbf{o}_i$, $i \in [t - p, t]$, and $D_{1:i}$, $i \in [1, l]$, denotes the first $i$ layers of the decoder. The process for obtaining $\mathbf{m}$ is detailed in Section 3.2.

**Navigational Action Prediction.** Finally, a head network is trained to predict both the action $\mathbf{a}_t$ and the waypoint distance vector $\mathbf{w}_t$. When feature selection is applied, this prediction process can be formulated as $\mathbf{a}_t, \mathbf{w}_t = h(\mathbf{x}_l)$, where $h$ represents the prediction head. In our implementation, $h$ consists of a 4-layer transformer followed by an MLP with a single hidden layer.

**Training Objective.** During training, we sample a sequence of visual images from the dataset to construct the observation $\mathbf{o}_{t-p:t}$. A goal image $\mathbf{o}_s$ is randomly selected for a valid prediction length, where $\mathbf{o}_s = \mathbf{o}_{t+d}$ and $d \in [t_{\min}, t_{\max}]$. The corresponding action sequence $\mathbf{a}_t^{\text{gt}} = \mathbf{a}_{t:t+d}$ and waypoint $\mathbf{w}_t^{\text{gt}}$ serve as ground truth. The objective of training is to maximize the likelihood of the predicted outputs aligning with the ground truth, formulated as:

$$\mathcal{L} = \mathbb{E}\left[\log p\left(\mathbf{a}_t^{\text{gt}} \mid \mathbf{a}_t\right) + \lambda \log p\left(\mathbf{w}_t^{\text{gt}} \mid \mathbf{w}_t\right)\right]. \tag{2}$$

## 3.2 Dynamic Sparse Feature Selection

End-to-end visual navigation models often operate as black boxes [4, 5, 60, 7], making it unclear which parts of an observation contribute most to action prediction. Understanding these key elements could enable targeted preprocessing to enhance model performance. This leads to a fundamental question: *should all pixels be treated equally?* Intuitively, the answer is no—emphasizing only relevant pixels improves robustness. In real-world scenarios, indiscriminate reliance on all pixels can lead to failures, especially when obstacles obstruct a robots camera. To address this, we introduce a feature selection approach based on the Gumbel-Softmax mechanism [61], dynamically prioritizing critical features. This improves performance and adaptability across diverse en-

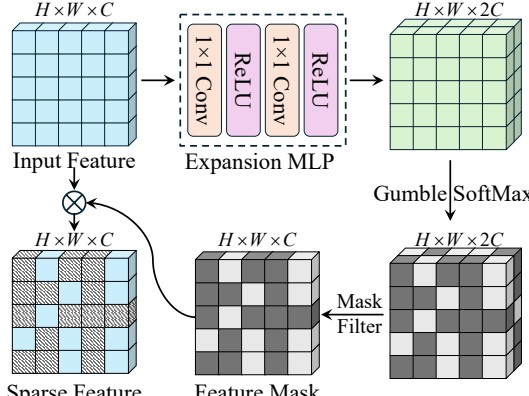

Figure 2: Architecture of the dynamic feature selector. A trainable MLP projects the input features to twice their original dimension. A pixel-wise Gumbel-Softmax operation is then applied to compute selection probabilities.

vironments and provides meaningful insights into the model's region of interest. The feature selector

functions as a classification network, assigning each pixel a probability score to generate masks for different input features. As shown in Figure 2, the feature selector $f(\cdot)$ takes encoded features as input and outputs corresponding masks as follows,

$$\mathbf{m}_i = f(\psi(\mathbf{o}_i)); \quad \mathbf{m}_s = f(\phi([\mathbf{o}_t; \mathbf{o}_s])) \in \mathbb{R}^{H \times W}. \tag{3}$$

Within the feature selector, the latent feature $\psi(\mathbf{o}_i)$ is projected into a higher-dimensional space:

$$\mathbf{Z}_i = MLP(\psi(\mathbf{o}_i)) \in \mathbb{R}^{H \times W \times C \times 2}, \tag{4}$$

where $MLP(\cdot)$ denotes a multi-layer perceptron. Here, $z_i^{n,c,k}$ represents the unnormalized log probability of the $k$-th category for the $n$-th pixel and $c$-th channel. To obtain the one-hot mask, we utilize the Gumble-SoftMax trick, which first adds a log term to each element and then conducts the SoftMax. The logarithm term is defined as:

$$g_i^{n,c,k} = -\log\left(-\log\left(u^{n,c,k}\right)\right); \quad u^{n,c,k} \sim U(0,1), \tag{5}$$

and the processed value in $\mathbf{Z}_i$ is $\bar{z}_i^{n,c,k} = z_i^{n,c,k} + g_i^{n,c,k}$. Then the SoftMax is applied on $\mathbf{Z}_i$ along the last dimension:

$$\hat{z}_i^{n,c,k} = \frac{\exp(\frac{\bar{z}_i^{n,c,k}}{\tau})}{\sum_{k'=1}^{2} \exp(\frac{\bar{z}_i^{n,c,k'}}{\tau})}, \quad k = 1, 2, \tag{6}$$

where $\tau$ is a temperature hyperparameter. At last, we manually define the last dimension of $\hat{\mathbf{Z}}_i$ as the generated mask, i.e. $m_i^{n,c} = \hat{z}_i^{n,c,2}$. The feature selector will gradually filter out the undesired features as the training continues.

Figure 3 presents the visualized input and its gradients that are processed through the feature selector. The spatial importance weights are visualized through saliency maps, where the attention mask is upsampled to match the input dimensions. The brightness intensity of each pixel in the visualization corresponds to its selection probability by the feature selection mechanism. The results indicate redundancy within the input data, while the navigation model does not specifically focus on the largest common object between the observation and the goal. This finding not only supports the feasibility of filtering pixels but also enhances the interpretability of the navigation process. After selection, we can utilize data sparselization techniques [62, 63] to save space.

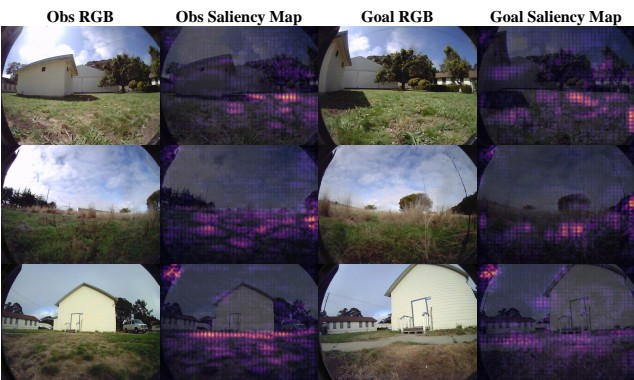

Figure 3: Visualization of saliency maps for observation and goal images.

### 3.3 Dynamic Transformer Layer Inference

### 3.3.1 Feature-Aware Early Exit Strategy

Transformer-based decoders are highly effective in visual navigation, leveraging long-range dependencies and flexible adaptation [64–67]. However, models like ViNT [5] employ a scene-agnostic decoder that activates all layers indiscriminately, disregarding scene complexity and task requirements. While beneficial for large-scale training, this approach imposes excessive computational demands on edge devices. We argue that activating every layer is often unnecessarysimilar to how humans selectively engage neurons for cognitive tasks [13, 68]. To address this, we propose a **dynamic** navigation decoder with an early-exit mechanism, allowing the model to halt computation based on scene complexity and navigation needs. By leveraging intermediate features for action prediction, this methodthe first to introduce early exiting in visual navigationeliminates redundant computations. Additionally, we enhance efficiency and robustness by integrating feature selection as

an initial step in the early-exit strategy. Our approach significantly reduces computational overhead while maintaining performance, making it well-suited for resource-constrained deployment.

Figure 1 outlines our early-exiting strategy workflow. DeeR-VLA [11] proposes a metric for robotic tasks, arguing that transformer layer features are inherently distinct. It uses an action consistency condition, measuring the difference in action outputs from an action head $h$:

$$|h(\mathbf{x}_i) - h(\mathbf{x}_{i-1})|_2 \le \eta_i, \quad \forall i \in \{1, 2, \ldots, l\}. \tag{7}$$

However, this still requires activating multiple layers, limiting computational savings. To improve efficiency, we introduce an aggressive early-exit strategy. When the L2 difference between the goal state and the current observation falls below a training-derived threshold (based on masked pixel counts), we bypass the transformer decoder entirely and compute actions directly from the encoded tokens and a prediction head.

### 3.3.2 Adaptive Threshold Optimization

To determine the optimal early-exit threshold, we employ Bayesian Optimization [69, 11] to iteratively search for the best value under given constraints. Specifically, we consider the predicted action $\mathbf{a}_t$ and waypoint $\mathbf{w}_t$ alongside their respective ground truth values, $\mathbf{a}_t^{\text{gt}}$ and $\mathbf{w}_t^{\text{gt}}$. Our objective is to maximize the cosine similarity between predictions and ground truth by optimizing the early-exiting thresholds, denoted as $\eta = \{\eta_1, \eta_2, \ldots, \eta_N\}$. Consequently, the objective function is formulated as:

$$\max_{\eta} J(\eta) = \frac{1}{T} \sum_{t=1}^{T} \left( \text{Sim}(\mathbf{a}_t, \mathbf{a}_t^{\text{gt}}; \eta) + \lambda \cdot \text{Sim}(\mathbf{w}_t, \mathbf{w}_t^{\text{gt}}; \eta) \right), \tag{8}$$

where $\text{Sim}(\mathbf{u}, \mathbf{v}; \eta)$ represents the cosine similarity between two vectors with a given early exit threshold $\eta$. $\lambda > 0$ is a hyperparameter that balances waypoint and action prediction, and $T$ is the total number of time steps in the task. To optimize this objective function, we introduce a penalty function $P(\eta)$ that enforces the required constraints. This function assigns a positive value when $\eta$ violates any constraint and remains zero otherwise. Incorporating this penalty into the optimization framework, we reformulate the problem as:

$$\max_{\eta} V(\eta) = J(\eta) - P(\eta), \tag{9}$$

where the penalty term, $P(\eta) = \sum_k \xi_k \cdot \max(0, g_k(\eta))$, captures the weighted sum of constraint violations. Here, $g_k(\eta)$ quantifies the extent to which the $k$-th constraint is violated, while $\xi_k$ represents its associated weight (remain constant). The specific constraints that the model must satisfy are outlined below.

**Inference Time Constraint**. Let $\text{Time}(\eta)$ denote the average inference time over the entire test set, where $\eta$ represents the early exit decision parameters. To ensure the efficiency of the network, we impose a constraint that the average inference time remains below a predefined threshold $T_{\max}$. Mathematically, this can be formulated as:

$$\text{Time}(\eta) = \frac{1}{T} \sum_{t=1}^{T} \text{Time}_t(\eta), \text{ s.t. } \text{Time}(\eta) \le \mathcal{T}_{\max}. \tag{10}$$

$T$ is the total number of test samples, $\mathcal{T}_{\max}$ is the maximum time, and $\text{Time}_t(\eta)$ denotes the inference time for the $t$-th sample under the given early exit strategy. This constraint ensures that the optimization selects an early exit criterion that balances computational efficiency, predictive performance, and real-time or application-specific latency requirements.

**GPU Memory Constraint**. To ensure efficient deployment under limited GPU resources, we define $\text{Mem}(\eta)$ as the GPU memory usage when applying the early exit strategy. Since memory consumption can fluctuate during inference, we consider the peak memory usage across all inference steps and enforce an upper bound constraint:

$$\text{Mem}(\eta) = \max_{t=1,\ldots,T} \text{Mem}_t(\eta), \text{ s.t. } \text{Mem}(\eta) \le G_{\max} \tag{11}$$

where $\text{Mem}_t(\eta)$ represents the memory consumption at time step $t$, and $G_{\max}$ denotes the maximum allowable GPU memory. This constraint ensures that Bayesian optimization selects an early exit criterion that not only improves efficiency but also maintains feasibility within hardware limitations.

Table 1: Quantitative Comparison on Benchmarks. We highlight our method with the colored font , the best and the second best value of each metric are reported with **bold** and underlined fonts, respectively.

| Dataset | Method | $\text{Sim}(\mathbf{a}_t, \mathbf{a}_t^{\text{gt}}) \uparrow$ | $\text{Sim}(\mathbf{w}_t, \mathbf{w}_t^{\text{gt}}) \uparrow$ | $\mathcal{L}_{action}\downarrow$ | $\mathcal{L}_{dist}\downarrow$ | FLOPs $(10^9)$ | Time(s/traj) | Memory (Gb) |
|---|---|---|---|---|---|---|---|---|
| | ViNT [5] | 94.49 | 96.20 | 0.285 | 6.94 | 4.37 | 0.379 | 19.07 |
| Recon [5] | NoMad [6] | - | **96.64** | 0.207 | 6.44 | 7.46 | 1.118 | 21.36 |
| | **Ours** | **94.92** | 96.53 | **0.191** | **6.26** | **1.93** | **0.228** | **13.35** |
| | ViNT [5] | 88.50 | 93.47 | 0.531 | 15.80 | 4.37 | 0.379 | 19.07 |
| Go-Stanford [4] | NoMad [6] | - | 93.51 | 0.507 | **12.93** | 7.46 | 1.118 | 21.36 |
| | **Ours** | **89.07** | **93.66** | **0.449** | 14.23 | **1.68** | **0.209** | **12.27** |
| | ViNT [5] | 89.66 | 93.16 | 0.686 | 10.95 | 4.37 | 0.379 | 19.07 |
| SACSoN [71] | NoMad [6] | - | 93.69 | 0.501 | 9.66 | 7.46 | 1.118 | 21.36 |
| | **Ours** | **90.54** | **93.72** | **0.493** | **9.62** | **1.68** | **0.209** | **12.27** |
| | ViNT [5] | 95.43 | 96.89 | 0.141 | 16.08 | 4.37 | 0.379 | 19.07 |
| SCAND [70] | NoMad [6] | - | **97.79** | **0.121** | **13.05** | 7.46 | 1.118 | 21.36 |
| | **Ours** | **96.85** | 97.03 | 0.130 | 14.41 | **1.93** | **0.228** | **13.25** |

**FLOPs Constraint**. One of the most critical considerations in optimizing the early exit strategy is controlling the computational cost, particularly in the transformer decoder. To achieve this, we define FLOPs($\eta$) as the average floating point operations (FLOPs) required per trajectory. Our goal is to ensure that the computational complexity remains within a predefined upper bound, $F_{\max}$, while maintaining the models performance. Formally, we express this constraint as:

$$\text{FLOPs}(\eta) = \frac{1}{T} \sum_{t=1}^{T} \text{FLOPs}_t(\eta) \text{ s.t. } \text{FLOPs}(\eta) \le F_{\max}. \tag{12}$$

Here, $\text{FLOPs}_t(\eta)$ represents the computational cost at each exit point $t$. By integrating this constraint into our Bayesian optimization framework, we explore the trade-off between computational efficiency and model accuracy, enabling us to identify the most effective early exit criteria within the computational limits.

## 4 Experiment

### 4.1 Experimental Setups

#### 4.1.1 Datasets

We evaluated our method in two experimental settings: benchmark datasets and simulated environments. For the benchmark datasets, we select four diverse datasets to assess the performance of our approach under various conditions. These include the Recon dataset [5], which provides medium-speed ($2m/s$) outdoor data to evaluate our method in real-world, dynamic outdoor settings, and the SCAND dataset [70], a medium-speed dataset featuring environmental interactions. Additionally, we include the Go-Stanford dataset [30] and the SACSoN dataset [71], representing low-speed ($0.5m/s$) and medium-speed indoor scenarios, respectively. These datasets allow us to test our method across environments with varying speed characteristics. All datasets are pre-processed following the ViNT method [5] to ensure consistency across experiments. For each dataset, we randomly split the data into training (80%) and testing (20%) sets. The implementation detail is illustrated in Section B in the Appendix.

#### 4.1.2 Evaluation Metrics

For the benchmark comparison, we report the cosine similarity between action angles and predicted waypoints, denoted as $\text{Sim}(\mathbf{a}_t, \mathbf{a}_t^{\text{gt}})$ and $\text{Sim}(\mathbf{w}_t, \mathbf{w}_t^{\text{gt}})$, respectively (in percentage). Since NoMad [6] only outputs waypoints through the diffusion process, we omit the action angle term for this model. To highlight the efficiency advantages of our approach, we report the FLOPs and memory usage of each model on the entire evaluation set. For inference, we measure the time required to predict a single trajectory for each model. Additionally, we report the loss values for the action vector and distance.

For the CARLA [72] simulation, we track the progress of our model-driven agent until it either reaches the target or encounters a collision. The success rate is calculated as the mean of the ratio of progress length to total trajectory length.

## 4.2 Evaluation in Real-world Benchmarks

Table 1 illustrates the performance of our method on RECON [5], Go-Stanford [4], SACSoN [71], and SCAND [70] datasets. We compare the performance with ViNT [5] and NoMad [6]. All models are trained from scratch. Our model saves about 58% FLOPs across all benchmarks compared to ViNT [5] while maintaining comparable accuracy. Figure 4 depicts the efficiency advantages of our model compared to ViNT [5]. Our approach achieves a 0.83% improvement in $\text{Sim}(\mathbf{a}_t, \mathbf{a}_t^{\text{gt}})$ and a 0.28% increase in $\text{Sim}(\mathbf{w}_t, \mathbf{w}_t^{\text{gt}})$ com-pared to ViNT [5] across four benchmarks.

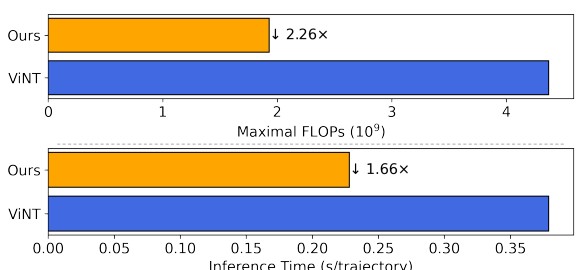

Figure 4: Efficiency comparison of **upper:** FLOPs, **bottom:** inference time between ours and ViNT on RE-CON dataset.

In terms of time efficiency, we save 0.16 seconds compared to ViNT [5] and 0.89 seconds compared to NoMaD [6]. Despite this, NoMaD [6], due to its diffusion refinement procedure, achieves an average performance that is 0.2% higher than ours. However, NoMaD [6] requires approximately four times the FLOPs of our method, making it less efficient. Notably, the average FLOPs of our dynamic model in RECON [5] and SCAND [70] are higher than those in SACSoN [71] and Go-Stanford [4]. One reason for this is that the former two datasets are from outdoor environments, while the latter two consist of indoor scenarios. The indoor datasets benefit from lower speeds, more controllable environments, and less complex lighting conditions. This finding also validates the assumption that for a more complex scene, activating more layers for accurate navigation.

## 4.3 Real-time Robotic Navigation in CARLA Simulation Environment

For the simulation, we first collected 200 trajectories of inline navigation data from CARLA [72] Town01 to fine-tune the pre-trained model. The data was gathered using an RGB cam-era and various sensors mounted on an autopilot agent operating at a fre-quency of 4Hz. To ensure consis-tency, we standardized the image size to $640 \times 480$ pixels with a $90°$ field of view (FOV). We evaluated our method across three distinct CARLA environments: Town02 (Scene A),

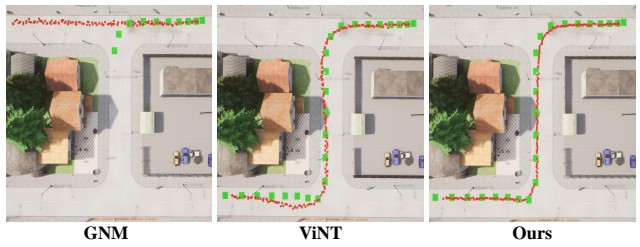

Figure 5: The simulation result in CARLA Town02 environ-ment. The green dots represent the discrete goals, and the red dots represent the predicted waypoints.

Town03 (Scene B), and Town10 (Scene C). Scene A, with a small-town layout and simple residential-commercial mix, represents an "easy task" for the agent. Scene B, a larger urban map with round-abouts and large junctions, is considered a "medium task." Scene C, a downtown area filled with skyscrapers, residential buildings, and parked cars, presents a highly dynamic and complex envi-ronment, making it a "hard task." For each scene, we collected 20 trajectories, which were used in the subsequent testing phase. The car was driven by a BehaviorAgent [72], maintaining a consis-tent maximum speed of 20 km/h across all environments. More details are illustrated in Appendix Section B.

Figure 5 illustrates the visualized navigation performance of baselines and ours. GNM [4] lacks enough generalization ability to achieve the target task. ViNT [5] and our method can successfully reach the target points. Note that the trajectory of ViNT [5] has some drift. This is due to ViNT [5] utilizing all features and decoder layers, potentially overfitting to training data and producing sub-

Table 3: Ablation study of the effectiveness of individual modules on the RECON dataset.

| Dynamic decoder | Feature selector | $Sim(\mathbf{a}_t, \mathbf{a}_t^{gt})$ | $Sim(\mathbf{w}_t, \mathbf{w}_t^{gt})$ | $\mathcal{L}_{action}$ | $\mathcal{L}_{dist}$ | FLOPs ($10^9$) | Time | Memory |
|---|---|---|---|---|---|---|---|---|
| Half layers | - | 91.05 | 93.28 | 0.332 | 7.53 | 2.61 | 0.306 | 17.48 |
| Half channel | - | 89.70 | 92.41 | 0.390 | 7.71 | 2.19 | 0.270 | 12.11 |
| - | - | 94.49 | 96.20 | 0.285 | 6.94 | 4.37 | 0.379 | 19.07 |
| ✓ | - | 93.68 | 95.42 | 0.274 | 7.08 | 2.41 | 0.251 | 16.49 |
| - | ✓ | 94.81 | 96.44 | 0.205 | 6.30 | 4.06 | 0.377 | 18.22 |
| ✓ | ✓ | **94.92** | **96.53** | **0.191** | **6.26** | **1.93** | **0.228** | **13.35** |

optimal trajectories. Our approach dynamically activates transformer layers and selectively filters features, resulting in superior trajectory performance.

Table 2 presents the success rate of different models on the CARLA [72] simulation. NoMad [6] is unable to achieve agile real-time simulation on our test platform due to the computationally intensive nature of its diffusion process. The results show that, although our model has higher FLOPs than GNM [4], its success rate shows a 38% improvement, demonstrating the effectiveness of our approach. Compared to ViNT [5], our method not only achieves comparable performance but also reduces FLOPs by more than a factor of two. Furthermore, as the simulation environment becomes more challenging (Scene A→Scene C), the FLOPs required by our model increase. This is because we

Table 2: The comparison of our model with baselines in the CARLA under various environments. The best and the second best values of each metric are reported with **bold** and underlined fonts, respectively.

| Environment | Model | Successful Rate | FLOPs ($10^9$) |
|---|---|---|---|
| Scene A | GNM [4] | 0.297 | 1.09 |
| | ViNT [5] | 0.724 | 4.37 |
| | Ours | **0.727** | 1.58 |
| Scene B | GNM [4] | 0.288 | 1.09 |
| | ViNT [5] | 0.659 | 4.37 |
| | Ours | **0.664** | 1.70 |
| Scene C | GNM [4] | 0.251 | 1.09 |
| | ViNT [5] | **0.589** | 4.37 |
| | Ours | 0.588 | 1.93 |

use a unified early exit metric across all three simulation environments. As the visual discrepancy between the observation and goal increases, our model needs more decoder blocks to extract contextual information effectively.

## 4.4 Ablation Study

**Ablation into Individual Modules:** Table 3 illustrates the effectiveness of our proposed module. The dynamic decoder column represents whether we are using early exit on the transformer decoder. The first row shows the result when we simply deactivate half of the decoder layers. Similarly, the second row presents the result when we deliberately reduce the hidden channel size from $C$ to $\frac{C}{2}$. Although these settings can improve efficiency, they usually lead to decreased performance and poor generalization (i.e., high accuracy on the training set but low accuracy on the testing set).. The rest of Table 3 elaborates that without the dynamic decoder, the efficiency does not vary too much compared to the baseline. Moreover, by using the feature selector, the performance will be better, and the efficiency will also be boosted. This is because our proposed feature selector sparsifies the features and stabilizes the early exit process.

**Ablation into Threshold Optimization:** Table 4 shows the different performances of whether we implement an extra Bayesian Optimization (BO) after training as DeeR-VLA [11]. Moreover, it reports the influence of whether we allow an early exit before the decoder. Results show that without BO, the early exit process can be impaired due to a suboptimal threshold. Besides, if we allow the early exit before the transformer decoder, although it can gain efficiency improvement, the over-

Table 4: Ablation study on whether using post-training Bayesian Optimization (BO) and allowing exit before the decoder. The best and the second best values of each metric are reported with **bold** and underlined fonts, respectively.

| BO | Pre-decoder Exit | $Sim(\mathbf{w}_t, \mathbf{w}_t^{gt})$ | Successful Rate | FLOPs ($10^9$) |
|---|---|---|---|---|
| - | - | 96.30 | 0.725 | 2.46 |
| - | ✓ | 96.22 | 0.719 | 2.27 |
| ✓ | - | **96.58** | **0.732** | 2.11 |
| ✓ | ✓ | 96.53 | 0.727 | **1.93** |

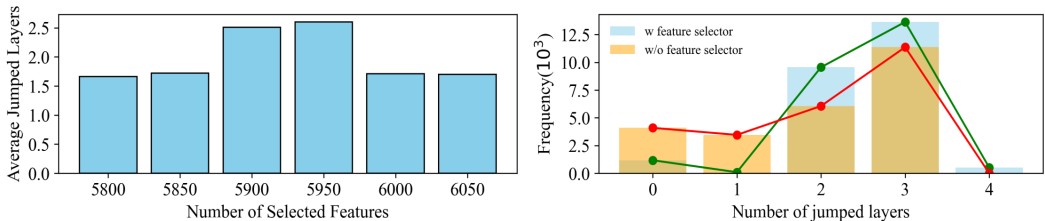

Figure 6: Visualized relationship between the number of selected features and the skipped layers (**Left**) and the frequency of different numbers of skipped layers in the context of with or without the feature selector (**right**).

all accuracy will slightly decrease. Therefore, such a technique ought to be a trade-off that requires careful design.

**Ablation into Feature Selection for Early Exit:** To assess the impact of our proposed feature selector on the early exit mechanism, we evaluate the model on the RECON [5] test set with a batch size of 1, both with and without the feature selector. For each sample, we record the MEAN value of the action difference between each layer (0-1,1-2,2-3). Moreover, we record the early exit index, referring to it as the number of skipped layers. In Figure 6, we observe that within a certain range of selected feature numbers, the average number of skipped layers remains high. This finding suggests that our feature selector helps determine an appropriate early exit threshold, thereby enhancing the frequency of early exiting. Additionally, Figure 6 shows that our proposed feature selector increases the frequency of 2-to-4 layer jumps, leading to improved efficiency. Therefore, by integrating the feature selector with early exit, our method achieves both more stable and more efficient performance.

# 5 Conclusion

In this work, we propose DynaNav, a novel, highly efficient visual navigation model. We first introduce a dynamic feature selector that filters observations and goals to extract robust, memory-efficient features. We also introduce feature-aware early exit criteria for the transformer decoders, using action consistency metrics optimized via Bayesian techniques. Our experimental results show a significant reduction in computational overhead compared to existing foundation navigation models while maintaining high performance across standard benchmarks and the CARLA simulation environment. The empirical evidence validates the effectiveness of our approach in achieving efficient and robust visual navigation.

To achieve optimal performance, our model requires an additional optimization process. Although the Bayesian optimization helps fine-tune the model and determine optimal thresholds, the added labor cost is non-negligible. Future work could involve implementing these optimization techniques concurrently with training to create a more streamlined end-to-end system. Furthermore, the proposed feature selection mechanism can be integrated with various CNN-based encoder models to improve their overall efficiency.

# 6 Acknowledgement

This research is supported by the National University of Singapore under the NUS College of Design and Engineering Industry-focused Ring-Fenced PhD Scholarship programme. Changhao Chen is funded by the Young Elite Scientist Sponsorship Program by CAST (No. YESS20220181) and the National Natural Science Foundation of China (NFSC) under the Grant Number 62573370.

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

# Appendices of
## *DynaNav: Dynamic Feature and Layer Selection for Efficient Visual Navigation*

## A Overview

Section B illustrates the hyperparameters and details for training and inference. Section C shows more experimental results. Section D illustrates more visualized results of the saliency heatmaps on observations and goals.

## B Implementation Details

For pre-training, we adopt the same parameter settings as ViNT [5] to ensure a fair comparison. A notable difference, however, is that we directly use the encoded features from EfficientNet-B0 [57] as tokens to the transformer layer, bypassing the MLP projection used by ViNT to reduce the dimensionality from 1280 to 512. This modification helps save computational resources and time. The input RGB images are resized to a resolution of $85 \times 64$, with a batch size of 256. Both ViNT and our models are trained for 100 epochs under these conditions. For fine-tuning, we set the learning rate to 1e-4 and train for 80 epochs, deactivating the warm-up stage during this process.

Table 5: Hyperparameter Settings.

| Hyperparameter | Value |
|---|---|
| **General** | |
| Train Epochs | 100 |
| Fine-tuning Epochs | 80 |
| Input Resolution | $85 \times 64$ |
| Training LR | 0.0005 |
| Fine-tuning LR | 0.0001 |
| Warmup Epochs | 3 |
| Optimizer | AdamW |
| LR Scheduler | Cosine Annealing |
| Batch Size | 256 |
| $\lambda$ in loss | 0.5 |
| **Backbone** | |
| Type | EfficientNet-b0 |
| Hidden Dim | 1280 |
| **Data** | |
| Length of past frames | 5 |
| Length of predicted waypoints | 5 |
| Max obs-goal distance(meter) | 20 |
| Min obs-goal distance(meter) | 0 |
| **Transformer Decoder** | |
| Number of layers | 4 |
| Attention Heads | 4 |
| **Bayesian Optimization** | |
| $\text{Sim}(\mathbf{a}_t, \mathbf{a}_t^{\text{gt}})$ Constraint | 0.950 |
| $\text{Sim}(\mathbf{w}_t, \mathbf{w}_t^{\text{gt}})$ Constraint | 0.960 |
| FLOPs Constraint ($10^9$) | 2.0 |
| Time Constraint (sec) | 0.3 |
| Memory Constraint (GB) | 14 |
| Optimization Epochs | 20 |
| Constraint of Masked Pixels (obs) | 2770 |
| Constraint of Masked Pixels (goal) | 3400 |
| $\xi$ | [0.8,0.5,1.0] |
| **CARLA Realted** | |
| Max Speed | 20km/h |
| Max Distance | 900m |
| Capture Frequency | 4Hz |

### B.1 Hyper-parameter Setting

The detailed hyper-parameter settings for our training and fine-tuning are in Table 5.

## C   More Experiment Result

### C.1   Results with Mamba Decoder

We also test the performance of the Mamba [73] block. Table 6 illustrates the results of substituting mamba. As the resolution of our feature maps is not large, the advantage of Mamba [73] can not be fully explored. On the other hand, the Mamba's [73] core computing structure - the state space model (SSM) and its high-order recursive calculations- causes its calculation volume to increase rapidly under high-dimensional features. Comparing the $\text{Sim}(\mathbf{w}_t, \mathbf{w}_t^{\text{gt}})$, the performance using Mamba [73] blocks is lower than ours. This may result from the fact that Mamba [73] has advantages on long sequences but may not be able to fully utilize its recursive modeling capabilities on short sequences. Transformer [58] is more suitable for capturing global dependencies. Even if the sequence is short, it can still use the self-attention mechanism to efficiently model the relationship between features.

During the CARLA [72] simulation, the models predictions are normalized to compute the necessary waypoint offsets. These offsets, combined with the vehicles current location, determine the target waypoint. A PID controller is employed to generate control signals based on the target waypoint. To ensure smooth trajectory generation within the CARLA environment, we use an image captured six timestamps ahead of the current observation as the objective, carefully tracking the models progress over each run. As ViNT [5] processes the waypoints in relative coordinates, represented as follows:

$$\mathbf{w}_t = (\mathbf{P}_{t+h} - \mathbf{P}_t) \otimes \mathbf{R}(\theta_t), \tag{13}$$

where $\mathbf{P}_{t+h}$ and $\mathbf{P}_t$ denote the position vectors of the goal and current points in world coordinates, respectively, and $\otimes$ indicates matrix multiplication. $\theta_t$ represents the vehicle's yaw, and $\mathbf{R}$ is the rotation matrix. Thus, the final target point is calculated as: $\mathbf{P}_{t+h} = \mathbf{P}_t + \hat{\mathbf{w}}_t \otimes \mathbf{R}(\theta_t)^{\top}$, where $\hat{\mathbf{w}}_t$ is the predicted waypoint offset.

Table 6: Quantitative Comparison on Benchmarks of ours and Mamba blocks.

| Dataset | Method | $\text{Sim}(\mathbf{w}_t, \mathbf{w}_t^{\text{gt}})$ | FLOPs($10^9$) |
|---|---|---|---|
| RECON [5] | Mamba [73] | 95.09 | 4.41 |
| | Ours | 96.53 | 1.93 |
| Go-Stanford [4] | Mamba [73] | 93.34 | 4.41 |
| | Ours | 93.66 | 1.68 |
| SacSoN [71] | Mamba [73] | 92.92 | 4.41 |
| | Ours | 93.72 | 1.68 |
| SCAND [70] | Mamba [73] | 97.28 | 4.41 |
| | Ours | 97.43 | 1.93 |

### C.2   Goal Image Viewpoint Investigation

In real-world scenarios, goal images often come from diverse sourcessuch as human-captured photosand may not exactly align with the agents ego-centric view. Understanding how such domain and viewpoint differences impact performance is critical.

To investigate this, we conducted additional experiments in CARLA under three challenging conditions:

- **Same location, different angle:** Goal image is taken from the same waypoint but with a camera orientation offset (within $\pm 15°$).

- **Nearby location, same angle:** Image is captured from a nearby position (within 5 meters), keeping the same orientation.
- **Nearby location, different angle:** Goal is from a nearby waypoint (within 5 meters) and a different orientation.

Table 7 illustrates the quantitative results. Our model exhibits graceful degradation as the domain gap increasesi.e., greater viewpoint or positional differences. However, it consistently outperforms the ViNT baseline across all settings, highlighting the robustness and generalization of our approach to goal images with moderate domain shifts.

Table 7: Comparison of our model and ViNT under varying goal image settings in CARLA Scene A

| Setting | Model | Success Rate | FLOPs ($\times 10^9$) |
|---|---|---|---|
| Same Position, Same Angle | ViNT | 0.724 | 4.37 |
| | Ours | **0.727** | **1.58** |
| Nerby Position, Same Angle | ViNT | 0.723 | 4.37 |
| | Ours | **0.725** | **1.58** |
| Same Position, Different Angle | ViNT | 0.694 | 4.37 |
| | Ours | **0.708** | **1.74** |
| Nearby Position, Different Angle | ViNT | 0.688 | 4.37 |
| | Ours | **0.691** | **1.79** |

## C.3 Timestep-wise Consistency

To investigate the potential timestep-wise inconsistency, we conducted an in-depth analysis on a 700-frame trajectory. We segmented the trajectory into 100-frame intervals and computed the average FLOPs and inference time for each segment, comparing our model against the baseline ViNT. As shown in Table 8, our model consistently reduces both computational cost and inference time across all intervals, while maintaining or improving action similarity $\mathrm{Sim}(\mathbf{a}_t, \mathbf{a}_t^{gt})$. In over 96% of the evaluated trajectories, our approach is more efficient than ViNT without any degradation in navigation accuracy.

These results demonstrate that, despite the dynamic nature of early exiting, our model exhibits stable, consistent, and efficient performance over time in practice.

Table 8: Timestep-wise results compared with ViNT

| | No. of Frame | 100 | 200 | 300 | 400 | 500 | 600 | 700 |
|---|---|---|---|---|---|---|---|---|
| ViNT | FLOPs ($10^9$) | 4.37 | 4.37 | 4.37 | 4.37 | 4.37 | 4.37 | 4.37 |
| | Avg Time (s) | 0.218 | 0.218 | 0.218 | 0.218 | 0.218 | 0.218 | 0.218 |
| | $\mathrm{Sim}(\mathbf{a}_t, \mathbf{a}_t^{gt})$ | 94.41 | 94.46 | 94.49 | 94.48 | 94.52 | 94.51 | 94.49 |
| Ours | FLOPs ($10^9$) | 2.02 | 1.95 | 1.92 | 1.96 | 1.85 | 1.93 | 1.93 |
| | Avg Time (s) | 0.194 | 0.190 | 0.189 | 0.191 | 0.185 | 0.190 | 0.190 |
| | $\mathrm{Sim}(\mathbf{a}_t, \mathbf{a}_t^{gt})$ | 94.76 | 94.88 | 94.92 | 94.90 | 94.92 | 94.92 | 94.92 |

## C.4 Additional Ablation Study of Constraints

Our adaptive threshold optimization incorporates three constraints designed to jointly enhance model efficiency across FLOPs, time, and memory usage. To evaluate their individual contributions, we conducted an ablation study by removing each constraint separately.

As shown in Table 9, enforcing the FLOPs constraint encourages more frequent layer skipping, effectively reducing inference time and memory consumption. However, removing either the time or memory constraint results in noticeable degradation across all efficiency metrics. This confirms that jointly optimizing all three constraints achieves the best overall performance and balanced resource utilization.

Table 9: Ablation Study On RECON Dataset

| Setting | FLOPs ($10^9$) | Time (s/traj) | Memory (GB) |
|---|---|---|---|
| Ours | 1.93 | 0.228 | 13.35 |
| w/o FLOPs constrain | 2.84 | 0.291 | 15.62 |
| w/o Time constrain | 2.55 | 0.273 | 15.09 |
| w/o Memory constrain | 2.16 | 0.255 | 14.75 |

## C.5 Study on Robustness

We conducted each navigation trajectory in CARLA 10 times to evaluate the robustness of our method. Table 10 reports the FLOPs, average execution time, and average successful rate for selected trajectories across these runs. As shown, the results exhibit minimal variance, indicating strong consistency and low randomness. This stability is attributed to the synergy between our feature selector and Bayesian optimization, which together enable adaptive yet reliable behavior across diverse scenarios.

Table 10: Results of different separation simulations.

| No. of Trajectory | 1 | 2 | 3 | 4 | 5 | 6 | 7 | 8 | 9 | 10 |
|---|---|---|---|---|---|---|---|---|---|---|
| FLOPs ($10^9$) | 1.91 | 1.93 | 1.92 | 1.91 | 1.90 | 1.92 | 1.94 | 1.93 | 1.93 | 1.91 |
| Avg Time (s) | 0.258 | 0.260 | 0.258 | 0.257 | 0.260 | 0.257 | 0.262 | 0.260 | 0.257 | 0.258 |
| Success Rate | 0.725 | 0.727 | 0.727 | 0.726 | 0.726 | 0.726 | 0.728 | 0.727 | 0.726 | 0.727 |

# D  More Visualizations

In this section, we added more visualizations of saliency maps. Such a saliency map helps to identify the interest area after being processed by our proposed feature selector. From Figure 7 to Figure 10, we can tell that the region of interest is not always located in the biggest common object between observation and goal images. The model "considers" more spatial information, which results in higher "attention" along the target direction. These findings support our claims in Section 1 that there is redundant information in the observation and goal. In other words, it proves the rationality of using the proposed feature selector to filter features.

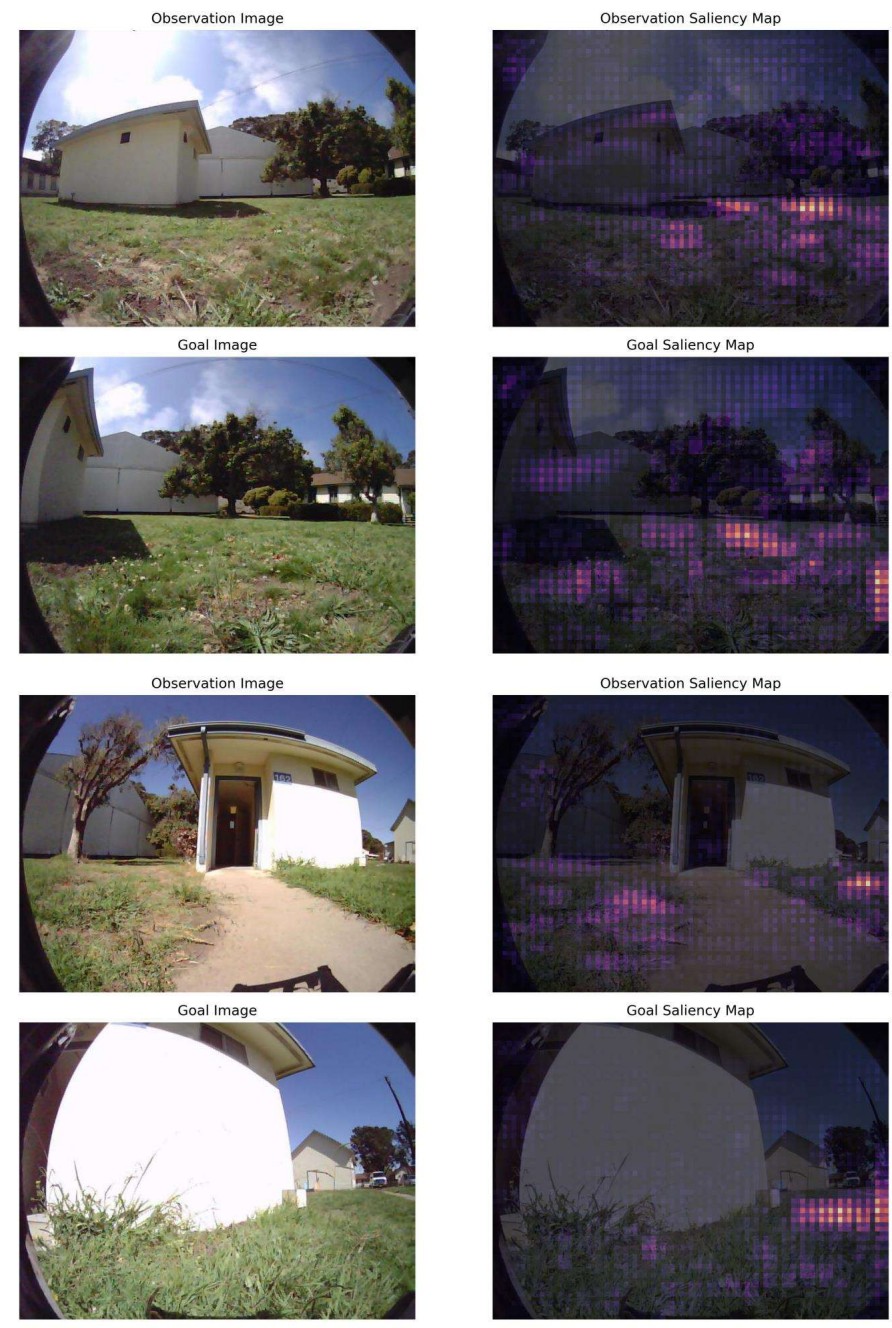

Figure 7: Salieny map of observation and goal images.

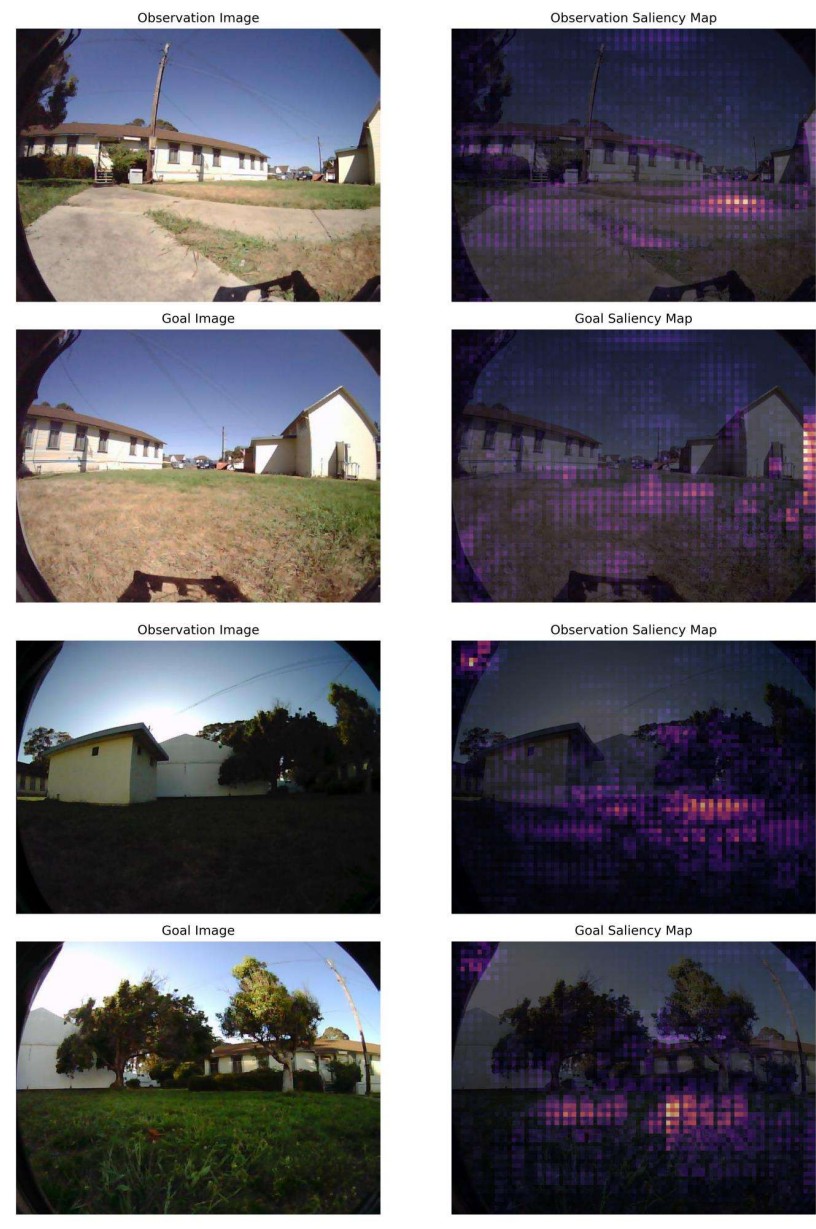

Figure 8: Salieny map of observation and goal images.

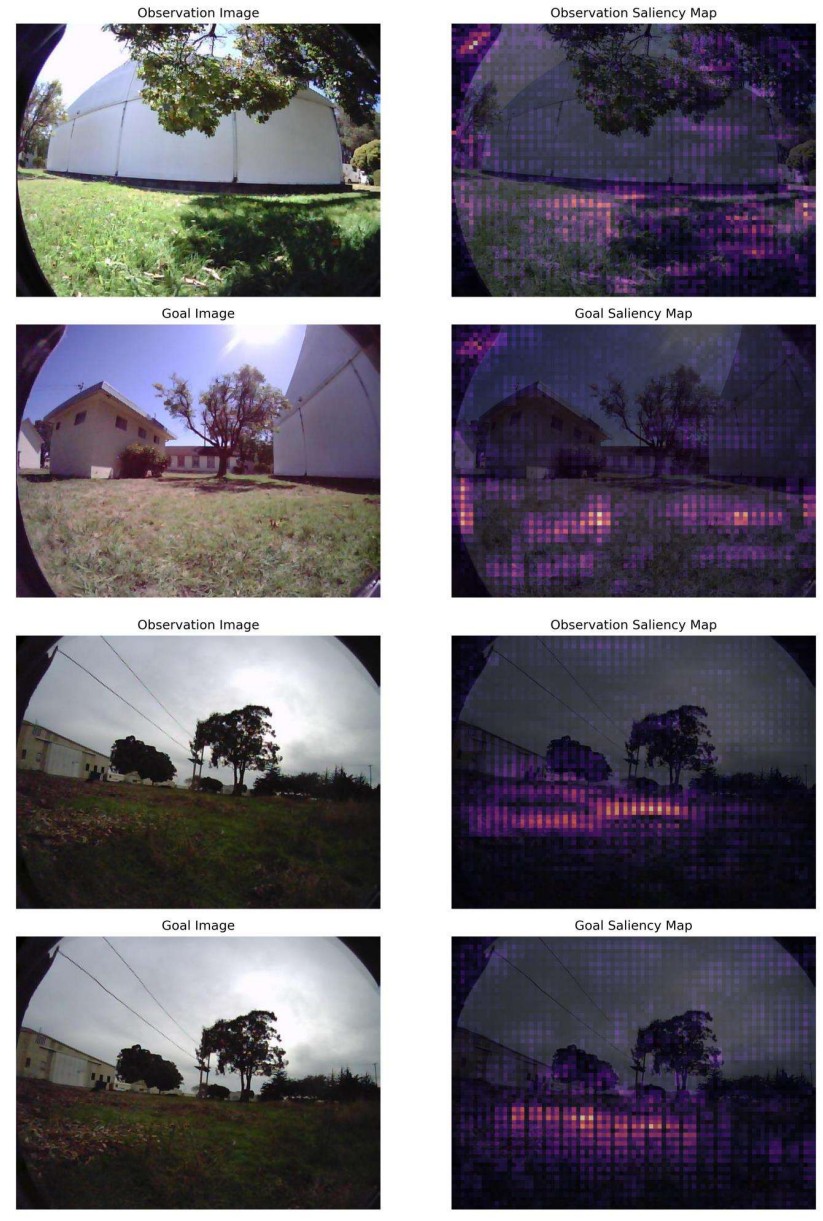

Figure 9: Salieny map of observation and goal images.

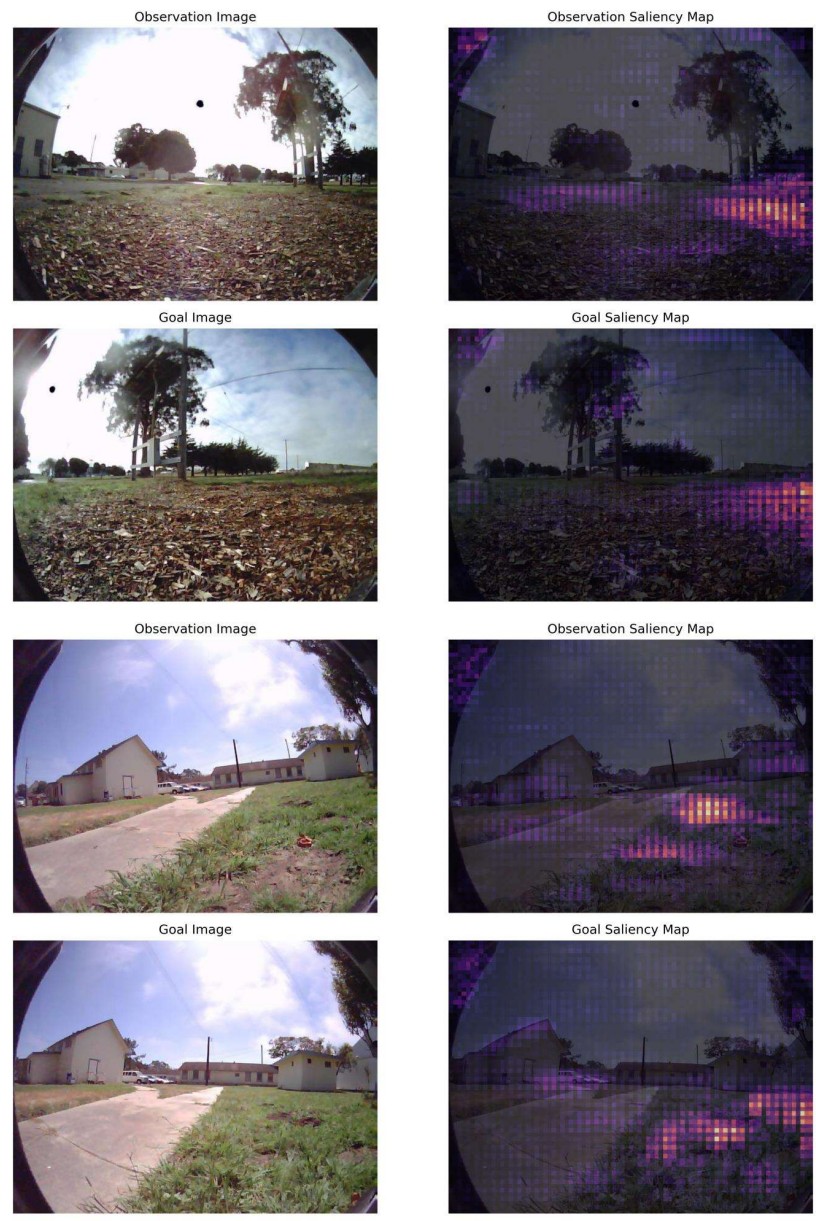

Figure 10: Salieny map of observation and goal images.

