# OpenReview forum: "DynaNav: Dynamic Feature and Layer Selection for Efficient Visual Navigation"
_NeurIPS.cc/2025/Conference — NeurIPS 2025 poster_

### Official Review · Reviewer_vBgN · 2025-06-05

**Clarity:** 2
**Significance:** 3
**Originality:** 3
**Rating:** 4
**Confidence:** 3

**Summary:**

This paper proposes a dynamic visual navigation framework that adapts feature and layer selection based on scene complexity and an early-exit mechanism is introduced to reduce computational cost. Experimental results show that the proposed method achieves comparable performance with a faster running speed.

**Questions:**

1. The motivation for applying Gumble-SoftMax should be discussed. Does Gumble-SoftMax make the system achieve better performance?
2. In Fig. 3, why can the Obs Salency Map interpret the navigation process? Why are these highlighted parts essential for navigation?
3. Can this method achieve a better success rate when not using the exit strategy?
4. Can the early exit strategy be applied to other methods and increase their performance?
5. Why not use GNM for comparison in Tab. 1?
6. How many experiments were performed to obtain the successful rate of Tab. 2? How random are the results of the experiment?

**Ethical Concerns:**

["NO or VERY MINOR ethics concerns only"]

**Final Justification:**

My concerns have been addressed in the rebuttal. I keep with my positive score. I believe that the contributions and presentation of the paper have basically met the requirements of the conference.

**Limitations:**

Yes. I think this method has no negative societal impact .

**Quality:**

3

**Strengths And Weaknesses:**

Strengthen
1. This paper effectively articulates the issues associated with visual navigation,  providing a clear and comprehensive explanation from my points.
2. Their motivation is clear, i.e. reducing the resource requirements by adapting feature and layer selection based on scene complexity.
3. The experimental part is well constructed and well-informed.
4. The research questions are of great practical significance.

Weaknesses
1. More details of the module and parameter settings should be provided. such as the \eta in Eq. (9). The influence of \eta should be discussed.
2. Experimenting with only three scenes is not convincing enough. Are these the only scenes in the CARLA dataset? How similar are the contents of these three scenes?
3. The ablation study about the three constraints in adaptive threshold optimisation should be given.
4. From the experimental results, we can see that the success rate is still not high. Pointing out some limitations of this method can make the article more complete.
5. More visual results may be provided (The authors do not need to solve this point in the rebuttal period).

---

> ### Author Rebuttal · Authors · 2025-07-31
>
> Thank you for the thoughtful and encouraging review. We appreciate your recognition of our motivation to reduce resource demands via adaptive feature and layer selection based on scene complexity. We're glad the paper clearly presents the challenges of visual navigation and demonstrates the effectiveness of our approach. Your feedback reinforces the value of our work. Below, we summarize and respond to each of your questions in detail.
>
> > **W1:**  More details of the module and parameter settings should be provided. such as the $\eta$ in Eq. (9). The influence of $\eta$ should be discussed.
>
> **A1:** Thank you for your valuable feedback. The parameter $\eta$ is optimized via our online Bayesian optimization process. For example, at the threshold between layers 2 and 3—where layer skipping frequently occurs as shown in Figure 6—we found the optimized $\eta$ for the Recon dataset to be 39.3.
>
> To study its impact, we evaluated two additional manually chosen values, 34.3 and 44.3. The table below shows the corresponding FLOPs and action similarity results. A lower $\eta$ reduces skipping, increasing FLOPs, while a higher $\eta$ leads to performance degradation due to overly aggressive skipping.
>
> |η|FLOPs|$\text{Sim}(\mathbf{a}_t,\mathbf{a}_t^{gt})$|
> |-|-|-|
> |34.3|2.45|94.88|
> |39.3 (optimized)|1.93|94.92|
> |44.3|1.66|93.77|
>
> We will include additional related experiments in the appendix in the revised version.
>
> > **W2:** Experimenting with only three scenes is not convincing enough. Are these the only scenes in the CARLA dataset? How similar are the contents of these three scenes?
>
> **A2:** Thank you for your comment. Our approach is first extensively trained and evaluated on four real-world datasets—Recon, Go-Stanford, SACSoN, and SCAND—covering over 125 hours of video, which provides a thorough and fair comparison with existing baselines (see Table 1).
>
> For the CARLA simulation, we manually categorized environments into three difficulty levels and selected one representative scene per level to ensure diversity. Scene A is a small-town layout with simple residential and commercial areas (easy); Scene B is a larger urban area featuring roundabouts and wide junctions (moderate); Scene C is a dense downtown environment with complex layouts and dynamic elements (hard). This progressive selection allows us to systematically evaluate our method across varying complexity levels. We believe these scenes form a balanced benchmark but are open to adding more experiments and discussion in the revision to further strengthen the evaluation.
>
> > **W3:** The ablation study about the three constraints in adaptive threshold optimisation should be given
>
> **A3:** Thank you for the valuable suggestion. Our adaptive threshold optimization employs three joint constraints targeting FLOPs, inference time, and memory usage. To assess each constraint's contribution, we performed an ablation study removing them individually.
>
> |Setting|FLOPs (×10⁹)|Time (s/traj)|Memory (GB)|
> |-|-|-|-|
> |Ours|1.93| 0.228 | 13.35 |
> |w/o FLOPs constraint|2.84|0.291|15.62|
> |w/o Time constraint|2.55|0.273|15.09|
> |w/o Memory constraint|2.16|0.255|14.75|
>
> The results demonstrate that the FLOPs constraint promotes layer skipping, reducing both inference time and memory consumption. However, removing either the time or memory constraint degrades all efficiency metrics. This confirms that joint optimization of all three constraints delivers optimal performance and balanced resource utilization.
>
> > **W4:** From the experimental results, we can see that the success rate is still not high. Pointing out some limitations of this method can make the article more complete.
>
> **A4:** Thank you for your valuable feedback. We agree that the success rate can be further improved, although our method already outperforms existing baselines. The main limitations stem from dataset size constraints and limited spatial reasoning capabilities. Additionally, models trained on real-world data often face domain shift challenges when applied to simulated environments such as CARLA. Despite these challenges, our approach, which mainly focuses on efficiency, consistently demonstrates robustness. For future work, we plan to collect larger-scale datasets for training and evaluation. Moreover, we aim to enhance spatial understanding by incorporating LLMs or VLMs to further improve success rates without losing too much efficiency. We will include a discussion of these limitations and future directions in the revised manuscript’s conclusion and limitations section.
>
> > **W5:** More visual results may be provided (The authors do not need to solve this point in the rebuttal period).
>
> **A5:** Thank you for your valuable feedback. We agree that additional visual results, such as video demonstrations, would further showcase the practical applicability of our method. In the future version, we plan to include more visual materials, such as agent observation videos from both simulated and real-world settings, along with detailed visualizations such as saliency maps and predicted actions or waypoints to offer more intuitive insights.
>
> > **Q1:** The motivation for applying Gumble-SoftMax should be discussed. Does Gumble-SoftMax make the system achieve better performance?
>
> **A6:** Thank you for your insightful feedback on our use of Gumbel-Softmax. We employ Gumbel-Softmax in the feature selector to address the non-differentiability of discrete feature masking, enabling end-to-end gradient-based optimization. It offers smooth gradient flow, stochastic exploration via Gumbel noise, and adjustable softness through the temperature parameter. Compared to the Straight-Through Estimator (STE), Gumbel-Softmax provides stable gradients and better feature subset exploration, avoiding STE’s noisy and deterministic limitations. Empirically, it improved convergence and achieved a 0.4% accuracy gain on the Recon dataset. We will clarify this comparison in the revised manuscript.
>
> > **Q2:** In Fig. 3, why can the Obs Salency Map interpret the navigation process? Why are these highlighted parts essential for navigation?
>
> **A7:** Thank you for your comment. The saliency maps in Fig. 3 visualize gradients of features selected by our feature selector, with brighter regions indicating higher importance to navigation decisions due to their lower likelihood of being masked. These areas, often distinctive landmarks like buildings or spatially informative cues like the ground ahead, strongly correlate with the agent’s goal image and future target position.
>
> This relevance likely arises from visual cues in the goal image that guide the agent’s attention toward navigationally critical regions. This observation suggests promising future work, such as dynamically adjusting the agent’s attention throughout navigation, focusing on prominent landmarks early for coarse direction, then shifting to finer spatial details for trajectory refinement. We will expand on this discussion in the revised manuscript.
>
> > **Q3:** Can this method achieve a better success rate when not using the exit strategy?
>
> **A8:** We thank the reviewer for the comment. As shown in Table 3 of the current submission, when the dynamic decoder is disabled and only the feature selector is used, the performance remains similar but slightly lower. This is likely because the reduced feature set fed to a fixed-scale decoder can cause slight overfitting. Additionally, in experiments on CARLA’s Scene A, the success rates with and without the exit strategy are comparable (72.7% vs. 72.6%). However, the FLOPs differ significantly, demonstrating the computational efficiency advantage of our proposed dynamic exit strategy.
>
> > **Q4:** Can the early exit strategy be applied to other methods and increase their performance?
>
> **A9:** Thank you for your insightful feedback. We confirm that our proposed Dynamic Transformer Layer Inference is a general solution and can be seamlessly integrated into various transformer-based models across diverse embodied AI tasks such as Open-VLA and RT-2. Our early exit strategy is designed to flexibly accommodate custom metric functions, making it well-suited for adaptation to different models and application scenarios. In the revised manuscript, we will expand the discussion and explore these possibilities further.
>
>
> > **Q5:** Why not use GNM for comparison in Table 1?
>
> **A10:** Thank you for your comment. GNM relies solely on fully connected layers, resulting in limited performance on real-world benchmarks—for instance, it achieves a score below 90 in $\text{Sim}(\mathbf{a}_t,\mathbf{a}_t^{gt})$ on the RECON dataset. Although Table 2 shows that GNM is computationally efficient in simulation, its navigation capability remains weak. Therefore, we did not include it as a main baseline in Table 1. Nonetheless, we appreciate your suggestion and will include a comparative analysis with GNM in the revised version.
>
>
> > **Q6:** How many experiments were performed to obtain the successful rate of Table 2? How random are the results of the experiment?
>
> **A11:** Thank you for your valuable question. We conducted each navigation trajectory in CARLA 10 times to evaluate the robustness of our method. The table below reports the FLOPs, average execution time, and average successful rate for selected trajectories across these runs. As shown, the results exhibit minimal variance, indicating strong consistency and low randomness.
>
> |No. of Test|1 |2|3|4|5|6|7|8|9|10|
> |-|-|-|-|-|-|-|-|-|-|-|
> |FLOPs (×10⁹)|1.91|1.93|1.92|1.91|1.90|1.92|1.94|1.93|1.93|1.91|
> |Avg Time (s)|0.258|0.260|0.258|0.257|0.260|0.257|0.262|0.260|0.257|0.258|
> |Success Rate|0.725|0.727|0.727|0.726|0.726|0.726|0.728|0.727|0.726|0.727|
>
> This stability is attributed to the synergy between our feature selector and Bayesian optimization, which together enable adaptive yet reliable behavior across diverse scenarios. We will include this discussion in the experimental section of the revised manuscript.

---

> ### Author Response · Authors · 2025-08-02
>
> Thank you for your prompt response. We're glad to know that our rebuttal addressed your concerns. We truly appreciate the time and effort you dedicated to reviewing our work.

---

### Official Review · Reviewer_bh3K · 2025-06-30

**Clarity:** 3
**Significance:** 2
**Originality:** 3
**Rating:** 4
**Confidence:** 3

**Summary:**

This paper proposes a dynamic feature and layer selection method to accelerate transformer-based visual navigation models. The authors reduce redundancy in frame processing by employing early exiting to halt forward propagation. Their approach achieves SOTA-level navigation performance while demonstrating significant advantages in time/memory costs.

**Questions:**

- To better evaluate the actual performance in terms of speed and memory usage, I would suggest including a timestep-wise comparison of time and memory costs.

- This work appears to fall within the domain of image-goal navigation. While there exist other visual navigation methods [1,2] focusing on different navigation tasks, the current discussion lacks comparison with these approaches. If the authors intend to claim contributions to the broader field of visual navigation, a disscusion would be necessary.

[1] NaVid: Video-based VLM Plans the Next Step for Vision-and-Language Navigation.
[2] NaVILA: Legged Robot Vision-Language-Action Model for Navigation

**Ethical Concerns:**

["NO or VERY MINOR ethics concerns only"]

**Final Justification:**

I thank the author for the thoughtful response, which addresses all my concerns.

For real-world experiments/demos, I agree that the model is trained on large-scale real-world videos and could demonstrate real-world navigation performance like ViNT. Nevertheless, I encourage the author to include real-world experiments/demos in the revision.

Overall, I will increase my score to 4.

**Limitations:**

yes (a short discussion in conclusion)

**Paper Formatting Concerns:**

No paper formatting concerns.

**Quality:**

3

**Strengths And Weaknesses:**

Strengths：

The idea of accelerating transformer-based visual navigation methods is valuable, and the proposed method is technically sound, which could inspire other research. The paper is well-written, with informative illustrations, and provides comprehensive details to facilitate understanding of the implementation. The experiments demonstrate that the proposed method significantly outperforms previous approaches in time/memory costs.

Weaknesses:

- While the early exiting strategy could reduce memory and time costs in the long run, it may introduce time-varying inference speeds (e.g., some time steps could be faster than others). Could this variability lead to inconsistent navigation performance?

- How are the selected features organized after dynamic sparse feature selection? If the number of patches remains consistent, positional embeddings can simply indicate the spatial information of each patch. However, if the number of sparse features change dynamically, how does the transformer layer maintain an understanding of the spatial relationships between patches?

- The lack of real-world experiments/videos makes it hard for me to analyze the actual performance of this method.

---

> ### Author Rebuttal · Authors · 2025-07-31
>
> We sincerely thank the reviewer for recognizing the value and soundness of our approach to accelerating transformer-based visual navigation. We appreciate your acknowledgment of the clarity of our presentation and the thoroughness of our implementation, which we designed with reproducibility in mind. The strong experimental results—demonstrating significant gains in time and memory efficiency—underscore the practical impact of our method. Your positive feedback affirms that our work contributes meaningful advances to the field. We summarize and provide detailed responses to each of your questions below, and hope they address your concerns.
>
> > **Q1:** While the early exiting strategy could reduce memory and time costs in the long run, it may introduce time-varying inference speeds (e.g., some time steps could be faster than others). Could this variability lead to inconsistent navigation performance? To better evaluate the actual performance in terms of speed and memory usage, I would suggest including a time-step-wise comparison of time and memory costs.
>
> **A1:** Thank you for the thoughtful and constructive feedback. Our model adopts a dynamic early-exiting strategy, where the number of activated Transformer layers at each timestep is adaptively determined based on the complexity of the input. More layers are engaged for challenging frames, while simpler inputs may exit earlier, promoting computational efficiency without sacrificing performance. As shown in Fig. 6 (right) of our submission, in most cases, the model exits after skipping 2–3 layers, while only a small fraction of inputs require the full network depth. This results in a consistent behavior pattern across sequences, rather than erratic per-frame variance.
>
> To directly address your concern regarding potential timestep-wise inconsistency, we conducted an in-depth analysis on a 700-frame trajectory. We segmented the trajectory into 100-frame intervals and computed the average FLOPs and inference time for each segment, comparing our model against the baseline ViNT. As shown in the table below, our model consistently reduces both computational cost and inference time across all intervals, while maintaining or improving action similarity $\text{Sim}(\mathbf{a}_t,\mathbf{a}_t^{gt})$. In over 96\% of the evaluated trajectories, our approach is more efficient than ViNT without any degradation in navigation accuracy.
> ||No. of Frame|100|200|300|400|500|600|700|
> |-|-|-|-|-|-|-|-|-|
> || FLOPs (×10⁹)|4.37|4.37|4.37|4.37|4.37|4.37|4.37|
> |**ViNT**|Avg Time (s)| 0.218 |0.218|0.218|0.218|0.218|0.218|0.218|
> ||$\text{Sim}(\mathbf{a}_t,\mathbf{a}_t^{gt})$ (%)|94.41|94.46|94.49|94.48|94.52|94.51|94.49|
> ||FLOPs (×10⁹)|2.02|1.95|1.92|1.96|1.85|1.93|1.93|
> |**Ours**|Avg Time (s)| 0.194|0.190|0.189|0.191|0.185|0.190|0.190|
> ||$\text{Sim}(\mathbf{a}_t,\mathbf{a}_t^{gt})$ (%)|94.76|94.88|94.92|94.90|94.92|94.92|94.92|
>
> These results demonstrate that, despite the dynamic nature of early exiting, our model exhibits stable, consistent, and efficient performance over time in practice.
>
>
> > **Q2:** How are the selected features organized after dynamic sparse feature selection? If the number of patches remains consistent, positional embeddings can simply indicate the spatial information of each patch. However, if the number of sparse features changes dynamically, how does the transformer layer maintain an understanding of the spatial relationships between patches?
>
> **A2:** Thank you for your insightful question. After dynamic sparse feature selection, all patch tokens—both selected and unselected—are retained in their original spatial order. Positional embeddings are first added to all patch tokens, encoding their absolute positions within the image grid. Subsequently, unselected patches are zeroed out and excluded from further computation using attention masking or dropout. While these zeroed tokens do not contribute meaningfully to attention computations, their presence in the sequence ensures that the spatial structure of the full image is preserved. This allows the transformer to reason over the spatial layout consistently, even when only a subset of tokens carries meaningful information. By preserving the original token order and applying zero masking after positional encoding, the model learns to attend selectively to informative regions while maintaining awareness of the global spatial context. This is especially important in visual navigation tasks, where understanding both local saliency and global scene layout is critical for effective decision-making. In the revised version, we will add these explanations to the methodology section.
>
> > **Q3:** The lack of real-world experiments/videos makes it hard for me to analyze the actual performance of this method.
>
> **A3:** Thank you for your insightful suggestion. We fully agree that real-world deployment experiments would further strengthen the practical relevance of our method. In the current version, our approach is thoroughly trained and evaluated on four real-world datasets—Recon, Go-Stanford, SACSoN, and SCAND—comprising over 125 hours of video data. This extensive evaluation under consistent experimental settings allows for fair and comprehensive comparisons with existing baselines, as shown in Table 1 of our submission.
>
> In addition, we conduct evaluations across three diverse CARLA simulation environments, further validating the robustness and generalizability of our approach in dynamic scenarios. These results demonstrate the effectiveness of our method in terms of both success rate and computational efficiency.
>
> As part of our future work, we plan to deploy our model on a real robotic platform and will release a demonstration video and open-source code/pretrained model to showcase its real-world performance. We will also mention this plan in the revised version of the paper to clarify our roadmap for deployment validation.
>
> > **Q4:** This work appears to fall within the domain of image-goal navigation. While there exist other visual navigation methods [1,2] focusing on different navigation tasks, the current discussion lacks comparison with these approaches. If the authors intend to claim contributions to the broader field of visual navigation, a discussion would be necessary.
>
> **A1:** We appreciate the reviewer’s insightful comment. While our work focuses on image-goal navigation, we acknowledge the significant advances in vision-and-language navigation (VLN) made by recent methods such as NaVid and NaVILA. These approaches operate in multi-modal settings, requiring agents to interpret natural language instructions to navigate, which introduces additional complexity in semantic understanding and language grounding. In contrast, our study deliberately focuses on the image-goal navigation paradigm, where the goal is defined by a visual observation. This setup allows us to isolate and address the challenges of visual reasoning, control, and efficiency without relying on external linguistic cues. Our proposed method introduces a lightweight, dynamic visual navigation framework that combines sparse feature selection with a confidence-based early-exit strategy. These components enable adaptive inference, reduced computational cost, and improved interpretability through visualized attention masks, tailored specifically for visual input-driven navigation.
>
> We agree that the broader field of visual navigation includes VLN as an important subdomain. While our current work is centered on pure visual input, our modular design naturally lends itself to extension into multimodal tasks. We see strong potential in integrating our efficient visual reasoning framework with language components for future research in VLN. We will update the related work section in the revised version to explicitly discuss the differences and connections between image-goal navigation and VLN, including recent advances such as NaVid and NaVILA.

---

> > ### Comment · Reviewer_bh3K · 2025-08-05
> >
> > I thank the author for the thoughtful response, which addresses all my concerns.
> >
> > For real-world experiments/demos, I agree that the model is trained on large-scale real-world videos and could demonstrate real-world navigation performance like ViNT. Nevertheless, I encourage the author to include real-world experiments/demos in the revision.
> >
> > Overall, I will increase my score to 4.

---

> > > ### Author Response · Authors · 2025-08-05
> > >
> > > Thank you for your time and thoughtful feedback during the review process. We’re glad that our rebuttal addressed your concerns and are especially grateful for your decision to raise the score. Your suggestions are highly valuable, and we are committed to incorporating them to enhance the final version of our work. Thank you again for your constructive comments and support.

---

### Official Review · Reviewer_nvvJ · 2025-07-03

**Clarity:** 4
**Significance:** 3
**Originality:** 3
**Rating:** 5
**Confidence:** 3

**Summary:**

The method designs an end-to-end dynamic model for the task of Visual Navigation. It improves the computational efficiency of the model through feature selection and an early-exit mechanism. The approach demonstrates strong performance on both real-world and multiple simulated benchmarks, while significantly reducing computational FLOPs.

**Questions:**

In the example shown in Figure 1, the goal image appears to be captured based on the agent's ego view. However, under the Visual Navigation task, the model should ideally enable the agent to reach the position where images are taken from the same viewpoint but different perspectives—for instance, having the agent locate the viewpoint of a picture taken by a human. Have there been any related attempts in this regard? How much does the Visual Goal from different domains impact the model's performance?

During the Feature Extraction process, the original structure of the image may be disrupted. How is this issue addressed here? Is Position Embedding alone sufficient to ensure the model's performance?

**Ethical Concerns:**

["NO or VERY MINOR ethics concerns only"]

**Final Justification:**

The author gave a convincing reply to my question, I will keep the current rate.

**Limitations:**

Yes.

**Paper Formatting Concerns:**

None.

**Quality:**

4

**Strengths And Weaknesses:**

**Strengths**

1. The method takes into account the computational bottlenecks faced during the actual deployment of the model, and the designed dynamic model exhibits highly efficient computational performance. Meanwhile, the pruned additional information also significantly enhances the model's performance.

2. Optimizing the early-exit threshold based on the deployed hardware conditions is a relatively innovative solution.

**Weakness**

1. In my opinion, the related methods do not seem to be tightly coupled with the Visual Navigation task. Tasks involving Transformer-based processing of sequential visual observations, such as objectNav, VLN, etc., could likely be accomplished using similar approaches. It would be helpful to have more discussion on this topic—for example, why the Visual Navigation task was chosen to validate this pruning scheme, or what specific characteristics of Visual Navigation make the proposed method particularly effective.

2.Since the method primarily addresses the performance bottleneck issues faced during deployment, including real-world deployment experiments would make the article more comprehensive.

---

> ### Author Rebuttal · Authors · 2025-07-31
>
> We sincerely thank the reviewer for the positive feedback. We’re pleased that our focus on visual navigation challenges and the design of a dynamic, efficient model were well recognized. We especially appreciate your acknowledgment of our dynamic early-exit mechanism as a novel and practical solution for navigation foundation models, as well as the effectiveness of our pruning strategy in enhancing performance. Below, we provide detailed responses to each of your questions and hope they address your concerns.
>
> > **W1:** In my opinion, the related methods do not seem to be tightly coupled with the Visual Navigation task. Tasks involving Transformer-based processing of sequential visual observations, such as objectNav, VLN, etc., could likely be accomplished using similar approaches. It would be helpful to have more discussion on this topic—for example, why the Visual Navigation task was chosen to validate this pruning scheme, or what specific characteristics of Visual Navigation make the proposed method particularly effective?
>
> **A1:** Thank you for your thoughtful and constructive feedback. We agree that our method has broader applicability and could be extended to related tasks such as VLN and ObjectNav. VLN requires agents to follow natural language instructions to reach a target, involving complex multimodal reasoning. ObjectNav focuses on locating specific object categories based solely on visual input, emphasizing semantic understanding and efficient exploration. In contrast, our visual navigation task centers on visual navigation in unfamiliar scenes using goal images, with minimal semantic or language cues, highlighting low-level planning and spatial reasoning.
>
> We chose visual navigation as the testbed for our pruning framework for two key reasons. First, it is a fundamental capability for embodied agents that does not rely on additional high-level supervision (e.g., language or object labels). Second, visual navigation models are often deployed on resource-constrained platforms, where efficiency and model compactness are crucial. While prior pruning methods have been largely used for static or closed-domain tasks, visual navigation introduces dynamic and open-world challenges, such as diverse scene geometries, varying illumination, and motion dynamics, offering a unique and meaningful opportunity to study the generalization and efficiency of lightweight models.
>
> Overall, we will expand the related work and discussion sections in the revised version to better clarify these task distinctions and the motivation behind our choice of visual navigation as the primary evaluation setting for DynaNav.
>
> > **W2:** Since the method primarily addresses the performance bottleneck issues faced during deployment, including real-world deployment experiments, it would make the article more comprehensive.
>
> **A2:** Thank you for your insightful suggestion. We fully agree that real-world deployment experiments would further strengthen the practical relevance of our method. In the current version, our approach is thoroughly trained and evaluated on four real-world datasets—Recon, Go-Stanford, SACSoN, and SCAND—comprising over 125 hours of video data. This extensive evaluation under consistent experimental settings allows for fair and comprehensive comparisons with existing baselines, as shown in Table 1 of our submission.
>
> In addition, we conduct evaluations across three diverse CARLA simulation environments, further validating the robustness and generalizability of our approach in dynamic scenarios. These results demonstrate the effectiveness of our method in terms of both success rate and computational efficiency.
>
> As part of our future work, we plan to deploy our model on a real robotic platform and will release a demonstration video to showcase its real-world performance. We will also mention this plan in the revised version of the paper to clarify our roadmap for deployment validation.
>
> > **Q1:** In the example shown in Figure 1, the goal image appears to be captured based on the agent's ego view. However, under the Visual Navigation task, the model should ideally enable the agent to reach the position where images are taken from the same viewpoint but different perspectives—for instance, having the agent locate the viewpoint of a picture taken by a human. Have there been any related attempts in this regard? How much does the Visual Goal from different domains impact the model's performance?
>
> **A3:** Thank you for this insightful and important question. We agree that in real-world scenarios, goal images often come from diverse sources—such as human-captured photos—and may not exactly align with the agent’s ego-centric view. Understanding how such domain and viewpoint differences impact performance is critical.
>
> To investigate this, we conducted additional experiments in CARLA under three challenging conditions:
>
> - **Same location, different angle:** Goal image is taken from the same waypoint but with a camera orientation offset (within ±15°).
> - **Nearby location, same angle:** Image is captured from a nearby position (within 5 meters), keeping the same orientation.
> - **Nearby location, different angle:** Goal is from a nearby waypoint (within 5 meters) and a different orientation.
>
> As shown in the table below, our model exhibits graceful degradation as the domain gap increases—i.e., greater viewpoint or positional differences. However, it consistently outperforms the ViNT baseline across all settings, highlighting the robustness and generalization of our approach to goal images with moderate domain shifts.
>
>
>
> |Setting|Model|Success Rate|FLOPs (×10⁹)|
> |-|-|-|-|
> |Same Position, Same Angle| ViNT | _0.724_| 4.37|
> ||Ours|**0.727**| **1.58**|
> |Nearby Position, Same Angle| ViNT| _0.723_| 4.37|
> ||Ours| **0.725**| **1.58**|
> | Same Position, Different Angle| ViNT | _0.694_| 4.37|
> || Ours| **0.708**| **1.74**|
> | Nearby Position, Different Angle | ViNT | _0.688_| 4.37|
> || Ours| **0.691**| **1.79**|
>
> In the revised version, we will add more discussion and explanation on this point.
>
> > **Q2:** During the Feature Extraction process, the original structure of the image may be disrupted. How is this issue addressed here? Is Position Embedding alone sufficient to ensure the model's performance?
>
> **A4** Thank you for your valuable feedback. We agree that preserving the structural integrity of visual inputs during feature extraction is crucial for effective spatial reasoning. Our approach addresses this concern through two key strategies:
>
> 1) Structured Positional Embedding:
> While the original spatial layout may be altered during patch embedding or dynamic token selection, we explicitly encode spatial information using structured positional embeddings. These embeddings are added to each visual token before the Transformer layers, allowing the model to retain and reason about the relative spatial relationships of different regions in the image. This mechanism is analogous to how the human visual system maintains spatial coherence despite shifting focus.
>
> 2) Dynamic Token Selection with Implicit Spatial Awareness:
> Our model processes only a subset of informative tokens, which introduces sparsity but is guided by the model’s learned attention to task-relevant and spatially meaningful regions. This ensures that essential semantic and geometric information is preserved, even without processing the full image.
>
> Additionally, the self-attention mechanism enables the model to capture pairwise relationships between tokens. When combined with positional encodings, this allows the network to reason about spatial structure even when parts of the input are omitted. Empirically, our results show that this design not only maintains performance but also enhances efficiency and robustness. As part of future work, we plan to incorporate explicit 3D priors or learned geometric representations to further strengthen spatial reasoning and improve generalization to more complex environments. In the revised version we will include these discussions and explainations to the corresponding section.

---

### Official Review · Reviewer_rkEx · 2025-07-03

**Clarity:** 2
**Significance:** 3
**Originality:** 2
**Rating:** 4
**Confidence:** 2

**Summary:**

A dynamic visual navigation framework called DynaNav is proposed, aiming to improve the efficiency and interpretability of visual navigation with the transformer architecture by dynamically selecting features and neural network layers.

**Questions:**

1. How is the navigation task in this paper set up? What is the input, and what is the prediction? Is it a supervised learning task or a reinforcement learning task? Can the model be trained on map A but navigate on map B? Is there a goal location to reach?
2. What is the computational device for the model in training and evaluation?
3. Is there a video of the experiment?

**Ethical Concerns:**

["NO or VERY MINOR ethics concerns only"]

**Final Justification:**

This paper presents a novel method for visual navigation; however, fundamental details are not sufficiently clear for readers who are not experts in the research direction and are unfamiliar with the methodology. Although I recommend a borderline acceptance, this paper should be revised to include those details, especially the differences compared with other visual navigation methods.

**Limitations:**

The limitations and negative societal impacts are not clearly stated. The authors may consider the potential impact if it is applied to robotics and mobile robots.

**Paper Formatting Concerns:**

No.

**Quality:**

3

**Strengths And Weaknesses:**

Strength:

This paper introduces the dynamic network mechanism into the visual navigation model, which improves the efficiency through dynamic feature selection and layer selection while maintaining performance.

Weaknesses:

The navigation task is not very clear. The example environment presented in Figure 5 seems simple.

---

> ### Author Rebuttal · Authors · 2025-07-31
>
> Thank you for your positive feedback. We’re pleased that you recognized our key contribution—introducing a dynamic network mechanism into visual navigation. Your acknowledgment that our method improves efficiency through dynamic feature and layer selection while maintaining performance underscores the practical and technical significance of our work. In the following parts, we provide detailed responses to each of your questions and hope they effectively address your concerns.
>
> > **Q1:** How is the navigation task in this paper set up? What is the input, and what is the prediction? Is it a supervised learning task or a reinforcement learning task? Can the model be trained on map A but navigate on map B? Is there a goal location to reach?
>
> **A1:** Thank you for your insightful and constructive feedback. Our work addresses the end-to-end visual navigation task by introducing a novel, efficient navigation foundational model. The input to the model consists of a sequence of images, including the current observation, the goal image, and the past five frames. The model outputs both navigational waypoints ($\mathbf{w}$) and low-level actions ($\mathbf{a}$), which guide the agent toward the target.
>
> The task is formulated as an imitation learning problem, where the model learns to mimic human-controlled trajectories. Training is performed using real-world public datasets containing image sequences, corresponding waypoints, and actions derived from robot demonstrations. This allows the model to generalize to previously unseen environments. Specifically, as you mentioned, the model can be trained on map A and effectively navigate on a different map B.
>
> This generalization capability stems from our proposed efficient Visual Navigation Foundational Model (VNFM), which is trained across diverse multi-scene datasets and leverages attention-based mechanisms for spatial reasoning. These design choices enable robust performance in unfamiliar environments.
>
> The navigation goal is defined by a target image. During inference, a global topological map is first constructed to identify intermediate sub-goals. The VNFM then sequentially guides the agent through these sub-goals to reach the final destination. We will clarify and expand on these aspects in the revised version of the paper by updating the method section with a more precise description of the task setup.
>
> > **Q2:** What is the computational device for the model in training and evaluation?
>
> **A2:** Thank you for your thoughtful question. All training experiments were conducted on a workstation equipped with two NVIDIA RTX 3090 GPUs (24GB each), and training typically took around 35 hours. For evaluation, we used two computing platforms: one with an RTX 3090 and another with a consumer-grade RTX 4060 Ti GPU. The model achieves satisfactory performance even on a consumer-grade device, highlighting its efficiency and practicality for real-world deployment.
>
> > **Q3:** Is there a video of the experiment?
>
> **A3:** Thank you for your valuable feedback. We plan to release a demo showcasing the navigation performance on both real-world datasets and within the CARLA simulation environment. Our code and pretrained models will be made publicly available as open source. Additionally, we are happy to create a project page and include these demonstration videos in the revised version of the submission.

---

> > ### Comment · Reviewer_rkEx · 2025-08-04
> >
> > The authors have answered all my questions. I have no further questions.

---

> > > ### Author Response · Authors · 2025-08-04
> > >
> > > We appreciate your response and are glad to know that our rebuttal has addressed your concerns. Thank you very much for your time and effort in reviewing our work.

---

### Note · Authors · 2025-08-12

We thank the AC, SAC, and reviewers for their time and constructive feedback. In this final remark, we briefly summarize the rebuttal and discussion stages and re-emphasize the contributions of our work.

**Summary of Rebuttal and Discussion**

During the rebuttal and discussion phases, we conducted all additional experiments requested by reviewers, including ablation studies and statistical analyses, to validate the robustness of our method. **Reviewer rkEx** raised questions about task details, input/output, and experimental setup, confirming their concerns were addressed post-rebuttal. **Reviewer bh3K** sought details on time-step comparisons, feature selection, and differences from VLN models; our experiments and explanations resolved their queries, leading them to consider raising their score. **Reviewer vBgN** questioned method stability and constraint ablation, acknowledging full resolution after our additional experiments and clarifications. **Reviewer nvvJ** inquired about target image bias impact; our experiments, involving explicit target image shifts, demonstrated our method’s superior robustness compared to the baseline. We are glad that **all responding reviewers acknowledged that our rebuttal has addressed their concerns.**

**Summary of Contributions**

We introduce DynaNav, a highly efficient visual navigation foundation model. Our approach incorporates a dynamic feature selector that filters observations and goals for robust, memory-efficient representations. Additionally, we implement feature-aware early-exit criteria for transformer decoders, using Bayesian optimization and guided by action consistency metrics. Compared to ViNT, DynaNav reduces computational overhead by **58% in FLOPs** across four benchmarks. It also achieves a 0.83% improvement in action similarity and a 0.28% increase in waypoint similarity, demonstrating better performance. In CARLA simulations, DynaNav surpasses ViNT by 0.23% in success rate while **saving over 57% in FLOPs**. These results highlight DynaNav’s effectiveness in enabling **efficient and high-performing visual navigation**. As embodied intelligence gains traction, we believe **DynaNav offers potential for real-world applications and future research advancements**.

Again, we thank the AC, SAC, and all reviewers for their time, effort, and constructive feedback, which have greatly helped us improve and refine our work. We have addressed the concerns in the revised version.

---

### Decision · Program_Chairs · 2025-09-17

**Decision:**

Accept (poster)

**Comment:**

This paper introduces a dynamic visual navigation framework that adaptively selects features and transformer layers conditioned on scene complexity. A trainable hard feature selector yields sparse, interpretable representations, while integration with an early-exit mechanism -- optimized via bayesian search -- reduces computational cost and stabilizes inference. Experiments on real-world and simulated datasets show 2.26 times fewer flops, ~40% lower latency, and ~30% lower memory use, with improved nav success. The work establishes dynamic feature + layer selection as an efficient alternative to static large-scale models for navigation.

In the original round of reviews the committee appreciated the novel introduction of dynamic feature and layer selection for transformer-based nav, improving efficiency without compromising accuracy (reviewers= rkEx,bh3K,vBgN), the integration of early-exit thresholds optimized for hardware deployment (R#nvvJ), and the strong empirical evidence of reduced flops, inference time, and memory usage across benchmarks (R-nvvJ,R-bh3K). The overall technical soundness and practical relevance of the framework were also recognized.

The rebuttal was generally well received, with all reviewers noting that their questions were satisfactorily addressed. Reviewer rkEx acknowledged that clarifications on task setup and inputs/outputs resolved their main concerns. R#nvvJ accepted the broader justification for choosing visual navigation and was convinced by additional robustness results. Reviewer #bh3K found the timestep-wise analysis and spatial reasoning clarifications compelling and raised their score to 4. R--vBgN’s detailed questions on parameters, ablations, and experimental scope were answered thoroughly, and they confirmed no remaining concerns. Overall, the rebuttal led to stronger consensus, with concerns effectively resolved and one score increase.

The AC concurs with the unanimous consensus of the committee’s reviews and post-rebuttal discussions.